# The Loss of Complex I in Renal Oncocytoma Is Associated with Defective Mitophagy Due to Lysosomal Dysfunction

**DOI:** 10.3390/ijms26157654

**Published:** 2025-08-07

**Authors:** Lin Lin, Neal Patel, Lucia Fernandez-del-Rio, Cristiane Benica, Blake Wilde, Eirini Christodoulou, Shinji Ohtake, Anhyo Jeong, Aboubacar Kaba, Nedas Matulionis, Randy Caliliw, Xiaowu Gai, Heather Christofk, David Shackelford, Brian Shuch

**Affiliations:** 1Department of Urology, UCLA, 405 Hilgard Avenue, Los Angeles, CA 90095, USA; 2Department of Urology, Weill Cornell Medicine, 530 East 70th Street, New York, NY 10065, USA; 3Department of Medicine, Endocrinology & the Metabolism Theme, UCLA, 405 Hilgard Avenue, Los Angeles, CA 90095, USA; 4Department of Biological Chemistry, Jonsson Comprehensive Cancer Center, UCLA, 615 Charles E Young Dr S, Los Angeles, CA 90095, USA; 5Center for Personalized Medicine, Children’s Hospital Los Angeles, 2100 W 3rd St, Los Angeles, CA 90057, USA; 6Department of Urology and Obstetrics and Gynecology, UCLA, 405 Hilgard Avenue, Los Angeles, CA 90095, USA; 7Broad Stem Cell Research Center, UCLA, 405 Hilgard Avenue, Los Angeles, CA 90095, USA; 8Division of Pulmonary and Critical Care Medicine and Jonsson Comprehensive Cancer Center, UCLA, 405 Hilgard Avenue, Los Angeles, CA 90095, USA

**Keywords:** renal oncocytoma, autophagy/mitophagy, lysosome, complex I, mitochondrial dysfunction, metabolism

## Abstract

Renal oncocytoma (RO) is a benign renal neoplasm characterized by dense accumulation of dysfunctional mitochondria possibly resulting from increased mitochondrial biogenesis and decreased mitophagy; however, the mechanisms controlling these mitochondrial changes are unclear. ROs harbor recurrent inactivating mutations in mitochondrial genes encoding the Electron Transport Chain (ETC) Complex I, and we hypothesize that Complex I loss in ROs directly impairs mitophagy. Our analysis of ROs and normal kidney (NK) tissues shows that a significant portion (8 out of 17) of ROs have mtDNA Complex I loss-of-function mutations with high variant allele frequency (>50%). ROs indeed exhibit reduced Complex I expression and activity. Analysis of the various steps of mitophagy pathway demonstrates that AMPK activation in ROs leads to induction of mitochondrial biogenesis, autophagy, and formation of autophagosomes. However, the subsequent steps involving lysosome biogenesis and function are defective, resulting in an overall inhibition of mitophagy. Inhibiting Complex I in a normal kidney cell line recapitulated the observed lysosomal and mitophagy defects. Our data suggest Complex I loss in RO results in defective mitophagy due to lysosomal loss and dysfunction.

## 1. Introduction

Renal oncocytoma (RO) is a type of benign kidney neoplasm characterized by dense accumulation of dysfunctional mitochondria [1]. ROs are commonly mistaken for renal cell carcinoma (RCC) owing to their similar patterns of contrast enhancement in CT or MRI [2]. Up to 30% of surgically resected small renal masses are benign, with oncocytoma being the most common one [3,4]. Although it is rare, some individuals develop large, multi-focal, bilateral, or symptomatic diseases, with some patients having renal parenchymal loss and renal failure, from a process known as renal oncocytosis [5]. To date, there is no reliable imaging modality to differentiate oncocytoma from malignant RCCs or effective medical treatment to inhibit tumor growth. Therefore, understanding the etiology and tumor biology of RO could prevent misdiagnosis and often unnecessary treatment, and improve patient outcomes.

ROs are found to have recurrent, inactivating nuclear and mitochondrial DNA mutations, often in genes encoding Electron Transport Chain (ETC) Complex I components [6,7]. Due to the lack of available in vitro or in vivo models, it is unknown how these alterations contribute to tumorigenesis. Complex I is a large transmembrane protein with 45 subunits, including 7 encoded by mitochondrial DNA and the remaining encoded by nuclear DNA [8]. The most commonly affected Complex I subunits in RO include MT-ND1, MT-ND5, MT-ND4, and MT-ND3, and they often present with high variant allele fractions (VAF) in oncocytomas (69–84%) and eosinophilic chromophobe RCCs (62%), and less frequently (<20%) in other renal malignancies [9].

The accumulation of mitochondria in RO is likely due to a combination of increased mitochondrial biogenesis and decreased mitophagy [6], a select form of autophagy through which cytoplasmic material is sequestered by autophagosomes and then targeted to autolysosomes for degradation. Autophagy is tightly regulated at a few critical steps. AMPK, a master metabolic sensor, activates autophagy in the setting of metabolic distress, such as with amino acid deprivation and energy deficits. Once AMPK is activated, it leads to assembly of the initiation complex containing phosphorylated ULK1 [10], activates formation of autophagosome, and stimulates mitochondrial and lysosomal biogenesis [11,12]. Next, cytoplasmic protein microtubule-associated protein 1 light chain 3 (LC3) is conjugated to phosphatidylethanolamine by enzymes like ATG7 and ATG3 to form LC3II. LC3II binds to autophagy receptors such as SQSTM1/p62, and it plays an important role in autophagosome maturation, autophagosome–lysosome fusion, and degradation of inner autophagosome membrane [13,14]. SQSTM1/p62 is normally degraded in autolysosome, and it accumulates when autophagy is inhibited [12].

In this study, we show that a significant portion (8 out of 17) of ROs have mtDNA Complex I loss-of-function mutations with high variant allele frequency (>50%). There is a significant loss of Complex I expression and activity in ROs, which is not entirely reliant on the specific Complex I mitochondrial mutation. ROs have increased mitochondrial content, which partially results from defective autophagy pathways. We show that increased AMPK activation in RO leads to mitochondrial biogenesis, induction of autophagy, and formation of autophagosomes. However, the subsequent steps involving lysosome biogenesis and function are defective. Inhibition of Complex I with rotenone in a normal kidney cell line recapitulated the characteristics of RO, including decreased lysosome content and defective mitophagy. These findings underscore the significance of lysosomal function and mitophagy regulation in the setting of RO.

## 2. Results

### 2.1. Prevalence of Complex I Mutations in Renal Oncocytomas

To study the effects of mitochondrial Complex I mutation in renal oncocytomas, we first sequenced mitochondrial DNA from 17 frozen ROs and their matched peripheral blood. We found that 13 (76.5%) have Complex I LoF mutations, including 8 (47.1%) with high variant allele frequency (VAF), defined as 50% and above (Figure 1A). Of the high VAF tumors, five had mutations in the ND6 subunit, two in ND4, and one in ND5. To correlate ETC Complexes’ activity with Complex I mutational status, we measured the oxygen consumption rate (OCR) of tissue homogenates using the Seahorse analyzer. There was a significant decrease in OCR from RO compared to normal kidney tissue (Figure 1B and Appendix AA). Concomitantly, there was a significant increase in Complex IV activity (Figure 1B and Appendix AA), but no change in Complex II in ROs (Appendix AB). Interestingly, when comparing the high VAF to the low VAF groups of ROs, we saw no difference in OCR (Figure 1C), suggesting that loss of Complex I activity in RO is independent of Complex I mutations.

To determine if Complex I expression loss in ROs is strictly associated with mitochondrial Complex I gene mutations, we analyzed the protein levels of Electron Transport Chain Complex components using Western blot. There was an obvious decrease in NDUFB8 protein levels, a component of Complex I, in ROs compared to normal kidneys (Figure 1D); however, the reduction was not directly associated with Complex I mutation status. Concurrently, there was an increase in Complexes III, IV, and V in ROs, suggesting a compensatory response [15]. Further, we showed that a nuclear-encoded Complex I subunit, NDUFA9, was also significantly decreased in oncocytomas (Figure 1E). Together these data demonstrated that Complex I activity is compromised in RO, a phenomenon that is not directly linked to mitochondrial Complex I mutation status within the mitochondrial genome.

### 2.2. Metabolic Changes in Oncocytoma Are Associated with Complex I Loss

To study the metabolic implications of Complex I loss, we analyzed gene expression using Nanostring nCounter metabolic panel. We showed that ROs have a distinct metabolic gene expression pattern compared to normal kidney tissues (Figure 2A). The most notable transcriptomic changes included upregulation of genes involved in AMPK signaling, mitochondrial biogenesis, and glutathione metabolism. ROs had increased expression of PRKAγ2, a subunit of AMPK, and Acetyl-CoA carboxylase β, a key enzyme in lipid biosynthesis and downstream effector of AMPK. Further, transcription factors associated with mitochondrial biogenesis, PPAR γ (proliferator-activated receptor γ) and PPAR γ-C1α (proliferator-activated receptor γ and coactivator 1 α), were also significantly upregulated in RO (Figure 2B). We therefore analyzed slides from resected RO for p-AMPK levels using immunohistochemistry (IHC), which demonstrated a remarkable increase in p-AMPK (Figure 2C).

We also observed a dysregulation of mediators of glutathione metabolism, including GCLC (Glutamate–Cysteine Ligase Catalytic Subunit) and BHMT-2 (betaine–homocysteine methyltransferase-2) (Figure 2D). We performed metabolite profiling using Liquid Chromatography–Tandem Mass Spectrometry (LC-MS) to investigate changes in glutathione metabolism (Figure 2E and Appendix AA). We observed significant increases in both oxidized and reduced forms of glutathione, GSH and GSSH, and its intermediary metabolite S-(1;2-Dicarboxyethyl)glutathione in RO, while its precursor, cystine, was significantly decreased in RO (Figure 2E and Appendix AA). Gene expression analysis also shows SOD1 and SOD3 (Superoxide Dismutase 1 and 3), key enzymes in Reactive Oxygen Species (ROS) processing, were significantly reduced in RO (Appendix AB,C), supporting the alterations in Glutathione metabolism pathways and ROS handling in ROs.

### 2.3. Complex I Loss in Oncocytoma Is Associated with Defects in Autophagy/Mitophagy

We further investigated the causes of accumulation of mitochondria in RO. Using IHC and Western blot, we found an increased abundance of TOM20, an outer mitochondrial membrane protein (Figure 3A,B). MIC60 and citrate synthase, mitochondrial inner membrane and matrix proteins, respectively, were both increased in RO (Figure 3B,C). To investigate the potential links between mitochondrial accumulation and autophagy, we investigated the levels of p62, an autophagy receptor that binds to LC3II and mediates autophagosome formation and maturation. We found that p62 was strongly accumulated in RO compared to its adjacent normal kidney (Figure 3D,E); however, LC3II protein levels were quite variable among RO specimens (Figure 3E), suggesting LC3II/p62 degradation, as well as LC3II conversion, may be altered in RO.

We next interrogated sequential steps of autophagy to identify the dysfunctional steps in the pathway. First, we used Western blot to investigate the phosphorylation of AMPK and its target protein ULK1, which is part of the autophagy initiation complex. The levels of p-AMPK at Thr172 and p-ULK1 were both significantly upregulated in RO, indicating enhanced induction of autophagy (Figure 4A) [16]. ATG7 plays a key role in LC3II conversion. A non-small-cell lung cancer model deficient in ATG7 was shown to transition to an oncocytic phenotype characterized by mitochondrial accumulation [17]. Interestingly, ATG7 expression in RO was significantly higher than normal kidney tissues by IHC and Western blot (Figure 4B,C). It was reported that in the setting of high GSH or Reactive Oxygen Species (ROS)-enriched environment, ATG7 function can be compromised by oxidation of its thio group, making it preferentially form a disulfide bond with itself or with its partner ATG3, instead of its traditional substrate, LC3. Therefore, we investigated if ATG7 function was compromised in oncocytoma by using non-reducing WB to determine if ATG7 was actively forming functional complexes. A non-reducing WB lacks β-mercaptoethanol and keeps the disulfide bond intact. As shown in Figure 4D, ATG7-LC3 complex band only existed in oncocytoma specimens, not in normal kidneys, indicating ATG7 was actively forming functional complexes with LC3. As expected, this complex was not visible on regular reducing WB because β-mercaptoethanol reduced the disulfide bond and broke up the ATG7-LC3 complex that would be no longer visible. These data demonstrated that ATG7 is highly expressed and remains functionally competent, suggesting that in RO, the autophagy pathway is intact up to autophagosome formation (Figure 4D).

### 2.4. Complex I Loss in Oncocytoma Is Associated with Lysosome Loss and Dysfunction

To evaluate the role of the lysosome in RO-associated autophagy, we first investigated the abundance of lysosomal proteins using Western blot and NK specimens. The classic lysosome marker, LAMP1 (lysosome-associated membrane protein 1), and lysosomal cysteine protease, cathepsin B, were both reduced (Figure 5A,B). It was previously reported that Complex I inhibition resulted in Golgi disassembly and LAMP2 retention; therefore, we quantified LAMP2 using Western blot [6]. We observed a change in the LAMP2 glycosylation pattern, with a drastic increase in the 70 kD band corresponding to the partially glycosylated LAMP2 (Figure 5B). IHC demonstrates an alteration of its subcellular distribution from a primarily apical location in NK to a primarily subnuclear location in RO (Figure 5C). These findings suggest an association between lysosome dysfunction and Complex I loss.

To further investigate how Complex I loss may directly affect autophagy in vitro, we treated YUNK1, a SV40-immortalized Yale University normal kidney 1 (YUNK1) cell line [18], with rotenone or DMSO for 24 h and quantified lysosomes using lysotracker staining. After treatment with 0.5 μM rotenone, we saw a significant reduction in the average number of lysosomes per cell area and average intensity of lysosomes (Figure 6A). To see if lysosome function is impaired, we also evaluated the lysosome acidification utilizing an assay based on hydrolysis of DQ-BSA. Surprisingly, despite the reduction of cathepsin B in RO, lysosomal acidification was not significantly altered by rotenone treatment, probably due to the relatively low level of acidification at baseline (Appendix A).

Next, we sought to determine whether loss of Complex I causes impairment of mitophagy. We stained rotenone-treated YUNK1 cells with p62, TOM20, and LAMP1. The co-localization of all three markers indicated the presence of mitolysosomes, which was subsequently used to quantify mitophagy flux (Figure 6B). Bafilomycin A1 targets the V-ATPase and prevents the passage of protons into the lysosomal lumen; thus, it is widely used as an inhibitor of lysosomal acidification, which allows for the quantification of all mitolysosomes formed within the entire duration of the culture. The mitophagy flux of YUNK1 was calculated as the fold change from treatment with rotenone or DMSO in the presence of bafilomycin compared to that without bafilomycin. We compared the mitophagy flux within 6 or 24 h of treatment and found that rotenone treatment mildly increased mitophagy in the first 6 h of treatment. At the 24 h time point, the DMSO group continued to have high mitophagy flux (fold change 2.2 from 6 h time point), while the rotenone group had significantly less mitophagy flux (fold change 1.3 from 6 h time point), indicating the inhibitory effector of rotenone on mitolysosome formation (Figure 6C). These findings suggest that rotenone decreases lysosome content and impairs mitophagy through Complex I inhibition.

## 3. Discussion

In this study, we demonstrated a universal loss of Complex I expression and activity in frozen ROs as compared to NK tissues. This universal loss of Complex I in RO is not entirely dependent on its LoF mutations in mitochondrial genes encoding Complex I subunits. Gene-expression changes revealed the activation of AMPK pathway in RO, a key regulator of mitochondrial biogenesis, which further led to the induction of autophagy. We subsequently mapped out the critical steps of the autophagy pathway and discovered that the key defect primarily resides in lysosomal loss and dysfunction, with the more proximal steps of autophagosome formation remaining intact. We confirmed these findings using an in vitro model of normal kidney cells treated with a Complex I inhibitor. Inhibition of Complex I caused loss of lysosome activity and inhibited mitophagy flux by halting the formation of mitolysosomes.

Mitochondrial respiratory chain Complex I is a large transmembrane protein which consists of 46 subunits, 7 of which are encoded by mtDNA (ND1-ND6 and ND4L). We detected a high frequency of Complex I LoF mutations in 8 of 17 oncocytomas, a rate comparable to other cohorts [9]. To evaluate the functional status of Complex I, we performed the first focused respirometry analysis of OCR evaluating the association with Complex I loss. We found that, in RO, there is a universal loss of Complex I activity compared to NK, independent of their mitochondrial mutational status. This suggests that the nine ROs without a high frequency of LoF mitochondrial mutations may contain other genetic alterations in the Complex I nuclear subunits or assembly factors, leading to a similar phenotype. Alternatively, other uncharacterized genetic alterations or the epigenetic regulation of mitochondrial Complex I gene expression and assembly may also explain the loss of Complex I activity. In fact, eight of our ROs contained missense Complex I mitochondrial mutations, and two tumors only had missense mutations, without any other LoF mutations in mtDNA-encoded Complex I genes. Another study also described a universal loss of Complex I enzymatic activity in ROs even in those without any detectable pathogenic mutations or those with only missense mutations in mitochondrial genes, thus further corroborating our hypothesis [19]. Future functional studies or computational modeling to characterize Complex I mitochondrial missense mutations or nuclear mutations will facilitate the understanding of Complex I loss in renal oncocytomas [20,21].

Autophagy is tightly regulated at a few critical steps. The activation of AMPK, a master metabolic sensor, leads to the assembly of the initiation complex, formation of the autophagosome, and mitochondrial and lysosomal biogenesis [10,22]. Next, LC3 is conjugated to phosphatidylethanolamine by ATG7 and ATG3 to form LC3II. Subsequently, LC3II binding to its receptors leads to autophagosome maturation, autophagosome–lysosome fusion, and degradation of the inner autophagosome membrane [12,23]. Defective autophagy in RO had been previously described, with one of the defects suspected to be at the conversion of unprocessed LC3I to the processed form, LC3II [6]. Accumulation of LC3-I and the absence of LC3II puncta or autophagosomes reassemble the molecular characteristic of *atg7* conditional knockout mice [11], thus leading us to believe that there might be defects in ATG7. Contrary to our assumption, we observed an upregulation of ATG7 in oncocytomas, with the non-reducing gel displaying ATG7-LC3 complex in RO but not in the normal kidney. Increased ATG7 remains functionally competent despite the presence of high levels of ROS and glutathione, suggesting that the steps leading to autophagosome formation remain intact in RO.

Our findings demonstrated a striking reduction in LAMP1 and cathepsin B in RO, and an alteration of LAMP2 glycosylation and subcellular distribution, indicating severe lysosomal impairment in ROs (Figure 7). In fact, the lysosome degradative function is so indispensable for autophagy and the endosomal cargo clearance pathway that it is considered to be essential for the survival of certain cell types. The most well-known example is lysosome dysfunction in neuronal cells, resulting in protein aggregates and damaged organelles that ultimately lead to adult-onset neurodegenerative diseases, such as Alzheimer Disease, Parkinson’s Disease, and frontotemporal dementia [24]. In oncocytomas, lysosomal impairment could be the result of decreased lysosome biogenesis, disruption and retention of lysosome components, lysosomal dysfunction, or any combination of these. The main transcriptional factor responsible for lysosome biogenesis is TFEB (transcription factor EB), which promotes the transcription of lysosome genes by binding to the consensus motif known as the CLEAR (coordinated lysosomal expression and regulation) sequence in the promoter regions of those genes [13,14]. TFEB activity is negatively regulated by mTORC1-mediated phosphorylation, which promotes the cytoplasmic localization and inhibits its nuclear translocation [25]. It was recently discovered that, under energetic stress, AMPK directly phosphorylates the serine residuals of folliculin-interacting protein (FNIP1) and suppresses the function of the folliculin–FNIP1 complex, which permits the nuclear translocation of TFEB and expression of lysosome biogenesis genes [26]. There could be defects or dysregulation at any part of the folliculin/FNIP1-mTOR-TFEB axis that prevented the biogenesis of lysosome, as seen with oncocytoma. Interestingly, TFEB and FNIP are both key mediators in unique forms of kidney cancer that often have an oncocytic phenotype (translocation RCC and Birt–Hogg–Dube (BHD) kidney cancer). Interestingly, an estimated 5% of BHD renal tumors have the exact same phenotype as a RO [27].

There are some limitations to this study. First, our controls are independent normal kidney tissues that are not adjacent to oncocytomas from the same patient, thus possibly increasing the variability of our controls. However, obtaining adjacent normal kidneys in oncocytoma specimens is no longer feasible in the current treatment era, given the technical nature of partial nephrectomy or enucleation performed for small renal masses. Second, due to the lack of established and tested in vitro oncocytoma models, we used 24 h rotenone treatment to induce inhibition of Complex I in an established model of normal renal cortex. Acute loss of Complex I may have different effects from chronic respiratory chain deficiency [29]. It is unclear how oncocytomas achieve lysosome loss or functional loss, and whether acute and chronic respiratory chain loss differ in regard to their lysosomal regulation. The development of RO cell lines with Complex I subunit knockout or knockdown will allow for a further interrogation of the mechanism of lysosome dysfunction. Despite being a benign disease, these tumors share similarities with other cancers, including subtypes of RCC and other cancers that have similar mitochondrial defects. Therefore, understanding the metabolic derangements and cellular adaptation of Complex I loss will lead to knowledge that can be applied to a wide range of diseases, and it will inform novel approaches to diagnosis and therapy.

## 4. Materials and Methods

### 4.1. Tumor Specimens

Frozen RO and normal kidney tissues were obtained from nephrectomy resections. All specimens were previously collected and de-identified. All frozen ROs and normal kidney specimens were obtained from surgical specimens from Yale University (IRB 0805003787) and UCLA (IRB 19-001174), respectively. Normal kidney tissues were obtained as radical nephrectomy specimens, collecting the uninvolved cortical tissue.

### 4.2. Cell Culture

First, 1 × 10^6^ cells were seeded in 10 cm^2^ plates using DMEM (Dulbecco’s Modified Eagle Medium) containing 1 g/L glucose supplemented with 2 mM glutamine, 1 mM sodium pyruvate, 10% FBS, and antibiotic–antimycotics.

### 4.3. Mitochondrial DNA Sequencing

The long-range PCR for the mitochondrial DNA (mtDNA) sequencing method was previously described [30,31]. Briefly, 100 ng of genomic tumor/patient DNA was used as template in a 50 μL TaKaRa LA Taq Hot Start Polymerase PCR reaction (TaKaRa Bio USA, San Jose, CA, USA). The DNA product was sheared (Covaris, LLC, Woburn, MA, USA), and library preparation and sequencing were performed using the KAPA Hyper Prep kit (KAPA Biosystems, Wilmington, MA, USA). Sequencing was then performed using NextSeq™ 500 (Illumina, San Diego, CA, USA) or MiSeqDx™ (Illumina) with 2 × 101 bp paired-end reads. The average sequencing coverage across the mitochondrial chromosome was 16,000× for all samples.

### 4.4. Protein Gel Electrophoresis SDS-PAGE

Normal kidney and RO tissue samples were homogenized in 0.5 mL MAS buffer (70 mM sucrose, 220 mM mannitol, 5 mM KH_2_PO_4_, 5 mM MgCl_2_, 1 mM EGTA, and 2 mM HEPES; pH 7.2) using a glass–Teflon Dounce homogenizer. The homogenate was centrifuged, and the supernatant was stored at −80 °C. Protein concentration was determined using the Pierce^TM^ BCA Protein Assay Kit (ThermoFisher, West Hills, CA, USA). A total of 20 μg of tissue homogenate was mixed with NuPAGE LDS sample buffer (ThermoFisher), either with (reducing conditions) or without (non-reducing conditions) β-mercaptoethanol. Samples were loaded into NuPAGE 4–12% Bis-Tris gels (ThermoFisher), and gel electrophoresis was performed in a mini-gel tank (ThermoFisher).

### 4.5. Immunoblotting

Proteins were transferred to a methanol-activated PVDF membrane in a Mini Trans-Blot cell (BioRad, Hercules, CA, USA) on ice. Blots were blocked in 3% BSA in PBS–Tween 20 (1 mL/L) for one hour and incubated with the primary antibody overnight. Primary antibodies used were Anti-ACTIN (Sigma, MAB1501, St. Louis, MO, USA); anti-AMPK (Cell Signaling, #2532, Danvers, MA, USA); anti-ATG7 (Novus Biologicals, #683906, Centennial, CO, USA); anti-cathepsin B (Cell Signaling, #31718); anti-citrate synthase (Abcam, ab96600, Cambridge, UK); Anti-LAMP1 (Cell Signaling, #9091); anti-LAMP2 (ThermoFisher, PA1-655); anti-LC3II (Cell Signaling, #3868); anti-MIC60 (Abcam, ab137057); anti-NDUFA9 (Abcam, ab14713); anti-p62 (Cell Signaling, #5114); anti-phospho AMPK Thr172 (Cell Signaling, #2535); anti-phospho S6 Ser240/244 (Cell Signaling, #2215); anti-phospho ULK1 Ser555 (Cell Signaling, #5869); anti-PROHIBITIN (Abcam, ab28172); anti-RAB7 (Cell Signaling, #9367); anti-S6 (Cell Signaling, #2217); anti-TOM20 (Sigma, WH0009804M1); anti-ULK1 (Cell Signaling, #8054); anti-Vinculin (Sigma, V9131); and total OXPHOS human WB Cocktail (Abcam; ab110411). Then, membranes were washed with PBS-T, incubated with the appropriate HRP-conjugated secondary antibody, and washed with PBS-T. Images were acquired in a ChemiDoc Imaging System (BioRad), and band densitometry quantified with Image Lab (BioRad).

### 4.6. Respirometry in Frozen Samples (RIFSs)

Respirometry in previously frozen samples was performed as previously described [32,33]. Briefly, 15 μg of tissue homogenate was loaded into a Seahorse microplate (Agilent, Santa Clara, CA, USA). Different substrates were injected into respective wells. For the exact methodology, see the Appendix A section. Rates were normalized by protein loaded and mitochondrial content. To determine mitochondrial mass, 3.75 μg of tissue homogenate was loaded into a clear-bottom 96-well microplate. Then, Mitotracker Deep Red FM (MTDR, ThermoFisher) was added to respective wells, and MTDR fluorescence was measured (λexcitation = 625 nm; λemission = 670 nm) to calculate mitochondrial content, defined as MTDR signal per microgram of protein.

### 4.7. Immunohistochemistry (IHC)

IHC was performed using Leica Bond RX processor, using Bond Polymer Refine Detection kit (Leica Biosystems, Cat# DS9800, Deer Park, IL, USA). Briefly, unstained slides underwent deparaffinization, antigen retrieval, peroxidase quenching, staining, and washing, following the manufactural protocol. Slides were incubated with primary antibodies, including p62 (Santa Cruz, #28359, Dallas, TX, USA), TOM20 (Cell Signaling #42406), ATG7 (Thermofisher, MAB 6608, Waltham, MA, USA), and p-AMPK (Cell Signaling #2535) for 60 min, and they were then incubated with Dakocytomation Envision System Labelled Polymer HRP anti rabbit (Agilent K4003) for 10 min. IHC images were acquired and processed with Aperio AT2 Scanner (Leica Biosystem).

### 4.8. Metabolite Extraction and LC-MS Analysis

Metabolites were extracted from 3 mg of frozen tissue using 80% MeOH + trifluoromethanesulfonate (internal standard). Metabolites were extracted using a Bead Mill Homogenizer (Fisherbrand), with tubes containing 1.4 mm ceramic beads. Metabolite-containing supernatant was dried over a continuous flow of N_2_, and then it was stored at −80 °C until it was analyzed by LC-MS. For each sample, 10 µL was run on a Vanquish (Thermo Scientific) UHPLC system with mobile phase A (20 mM ammonium carbonate, pH 9.7) and mobile phase B (100% acetonitrile). Each run was conducted using a flow rate of 150 µL/min on a SeQuant ZIC-pHILIC Polymeric column (2.1 × 150 mm 5 μm, EMD Millipore) and a temperature of 35 °C. Metabolites were separated into two linear gradients and one stable phase: (1) 80% B to 20% B, 20 min; (2) 20% B to 80% B, 30 s; and (3) 80% B, 7.5 min. Masses were determined using a Q-Exactive (Thermo Scientific) mass analyzer running in polarity-switching mode. Files were centroided, converted to positive and negative mzXML files (msconvert from ProteoWizard), and analyzed by MZmine2 using a built-in Automated Data Analysis Pipeline (ADAP). ADAP wavelet algorithm was used to detect peaks, which were then aligned, gap-filled, and assigned identities (+/−15 ppm) and retention times (+/−0.5 min). Peaks were assigned using an in-house database of known metabolites with associated retention times. Peak boundaries were manually curated and quantified by the area-under-the-curve integration, using an internal standard in the buffer to normalize samples.

### 4.9. RNA Extraction and Gene Expression Profiling

Total RNA was extracted using Qiagen RSC simplyRNA Tissue kit (Catalogue #AS1340, Promega Corp., Madison, WI, USA). The yield of RNA was determined using a NanoDrop ND-2000 spectrophotometer (NanoDrop Technologies, Wilmington, DE, USA) and QuantiFluor RNA System (Catalogue #E3310, Promega Corp, Madison, WI, USA). A total of 200 ng of RNA was analyzed using the NanoString nCounter Metabolic Pathways Panel (NanoString, Seattle, WA, USA).

### 4.10. Mitophagy Flux Quantification by Immunofluorescence

Cells were grown in a 96-well optical-bottom plate (themofisher) and cultured in the presence or absence of 0.5 μM Rotenone, Bafilomycin 0.1 μM, or DMSO 0.5 μM as control for 6 or 24 h, as indicated. Cells were fixed with 4% paraformaldehyde in phosphate-buffered saline (PBS) for 20 min at RT, permeabilized (0.1% Triton X-100, 0.05% sodium deoxycholate in PBS), blocked, and stained with primary and secondary antibodies in blocking solution (5% donkey serum). Antibodies used were SQSTM1/p62 (Cell Signaling #5114S); LAMP1 (Cell Signaling #51774), and TOM20 Alexa Fluor 647 Conjugate (Abcam ab209606). Cells were also co-stained with 1 ug/mL DAPI (nuclei) for further analysis, based on cellular mask. Imaging was performed in triplicates, using confocal mode and Z-stacks of 1 μm (6 μm total) with the ImageXpress Molecular Device system (40× water objective). Excitation and emission filters used for the combination of dyes were DAPI (ex. 360–400; em. 410–480), p62 (ex. 460–490; em. 500–550), LAMP1 (ex. 560–580; em. 590–640), and TOM20 (ex. 630–650; em. 640–680). Analysis was performed with Molecular Device (Harmony 4.1) software by masking whole cells using Gaussian-filtered TOM20 and Hoechst staining. Only whole cells were further analyzed. Inside each cell, p62 and LAMP1 were segmented into dots, while mitochondria were masked for the whole network. Total dots were counted per cell, as well as dots localizing together in the mitochondrial mask (also used to calculate mitochondria area per cell).

## Figures and Tables

**Figure 1 ijms-26-07654-f001:**
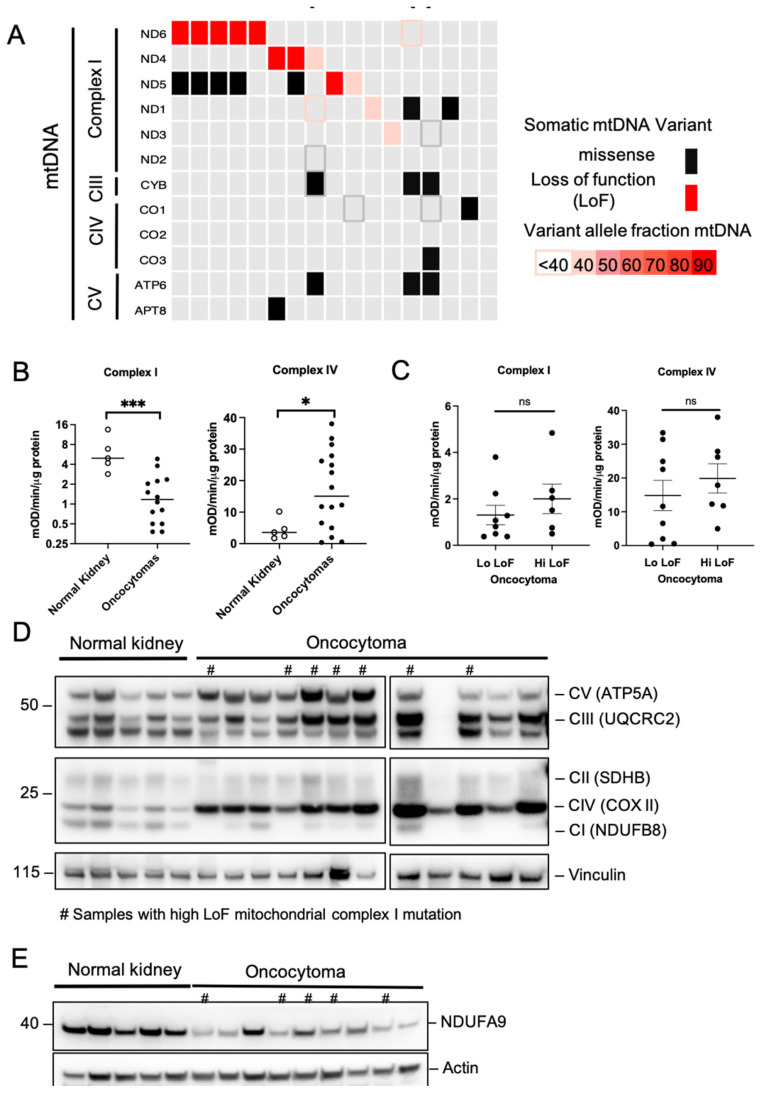
Complex I loss in renal oncocytomas is independent of the loss-of-function mutations in their mDNA. (**A**) Summary of somatic mitochondrial mutations identified. Each column represents a tumor sample, and each row represents the gene analyzed. Red boxes represent loss-of-function mutation, and black boxes represent missense mutations. The darker the color, the higher the variant allele frequency. Short bars (-) indicate samples lacking whole blood. (**B**) Complex I and Complex IV OCR from frozen oncocytoma and normal kidney lysates. (**C**) Complex I and Complex IV OCR from subgroups of low VAF and high VAF oncocytomas and normal kidney lysates. ns: non-significant. (**D**) Western blotting of the five complexes from frozen oncocytoma and normal kidney lysates. Pound sign (#) indicates samples with high VAF LoF Complex I mutations. (**E**) Western blotting of the NDUFA9, nuclear subunit of Complex I, from frozen oncocytoma and normal kidney lysates. ns: non-significant; * *p* < 0.05; *** *p* < 0.001.

**Figure 2 ijms-26-07654-f002:**
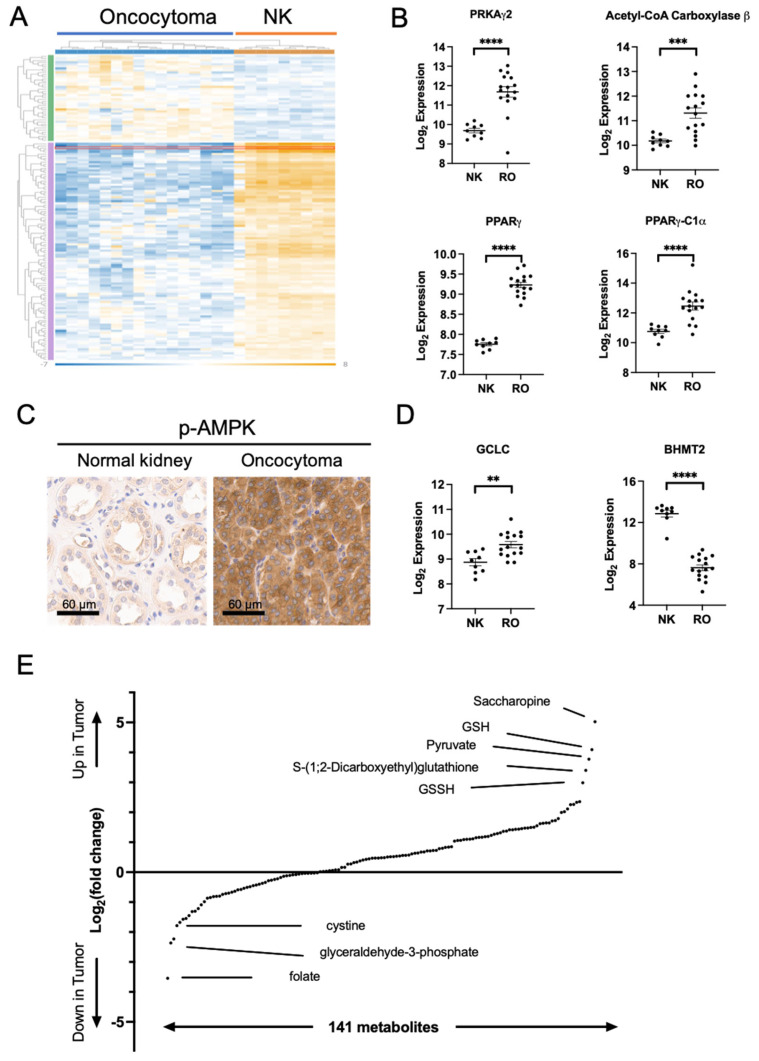
Activation of AMPK pathway and enrichment of GSH-GSSG in renal oncocytomas. (**A**) Gene expression profiles of RO and NKs using Nanostring nCounter metabolic panel. Blue: Oncocytoma, Yellow: normal kidney, green: over-expressed metabolic genes in RO, purple: over-expressed genes in normal kidney. (**B**) Gene expression of select enzymes downstream of AMPK. PPAR γ, proliferator-activated receptor γ; PPAR γ-C1⍺: proliferator-activated receptor γ and coactivator 1 alpha. (**C**) IHC of p-AMPK in normal kidney and renal oncocytoma. (**D**) Gene expression of select enzymes in the glutathione pathway. (**E**) Metabolite profiling using LC-MS. GCLC, Glutamate–Cysteine Ligase Catalytic Subunit; BHMT2: betaine–homocysteine methyltransferase-2. ** *p* < 0.005; *** *p* < 0.001; **** *p* < 0.0001.

**Figure 3 ijms-26-07654-f003:**
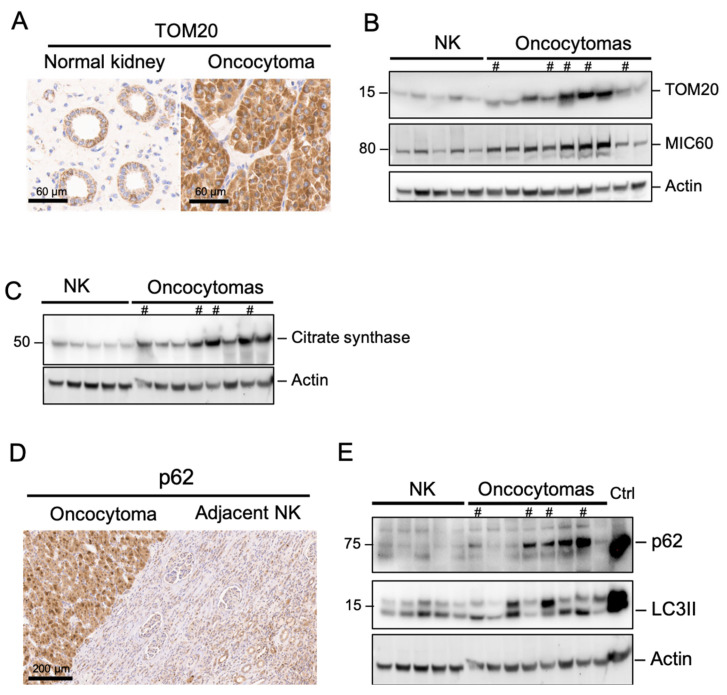
Accumulation of mitochondrial content and p62 suggests mitophagy defects in renal oncocytomas. (**A**) IHC of TOM20 in normal kidney and RO. (**B**,**C**) Western blot of mitochondrial markers. (**D**) IHC of p62 in oncocytoma and its adjacent normal kidney. (**E**) Western blot of autophagy flux markers, p62 and LC3II. # Samples with a high frequency of LoF Complex I mutations. The same loading control was used for Figure 3C,E. The same loading control was used for Figure 1E and Figure 3B since the same membrane was used.

**Figure 4 ijms-26-07654-f004:**
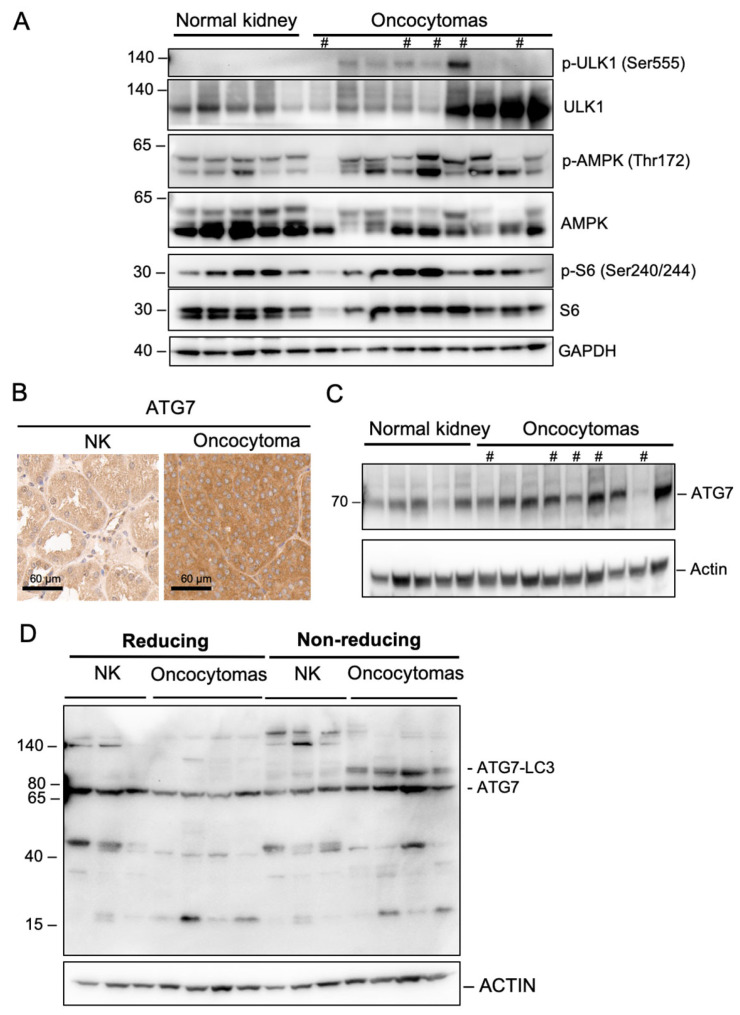
Autophagy pathway is intact up to autophagosome formation in oncocytomas. (**A**) Western blot of proteins associated with autophagy induction in frozen normal kidney and oncocytomas. (**B**) ATG expression in NK and RO using IHC. (**C**) Western blot of ATG7 expression in frozen NK and RO. (**D**) Non-reducing gel showing ATG7-LC3 complex formation in oncocytomas but not in normal kidney. # Samples with high frequency of LoF Complex I mutations. The same loading control was used for Figure 1E, Figure 3B and Figure 4C since the same membrane was used for immunoblotting.

**Figure 5 ijms-26-07654-f005:**
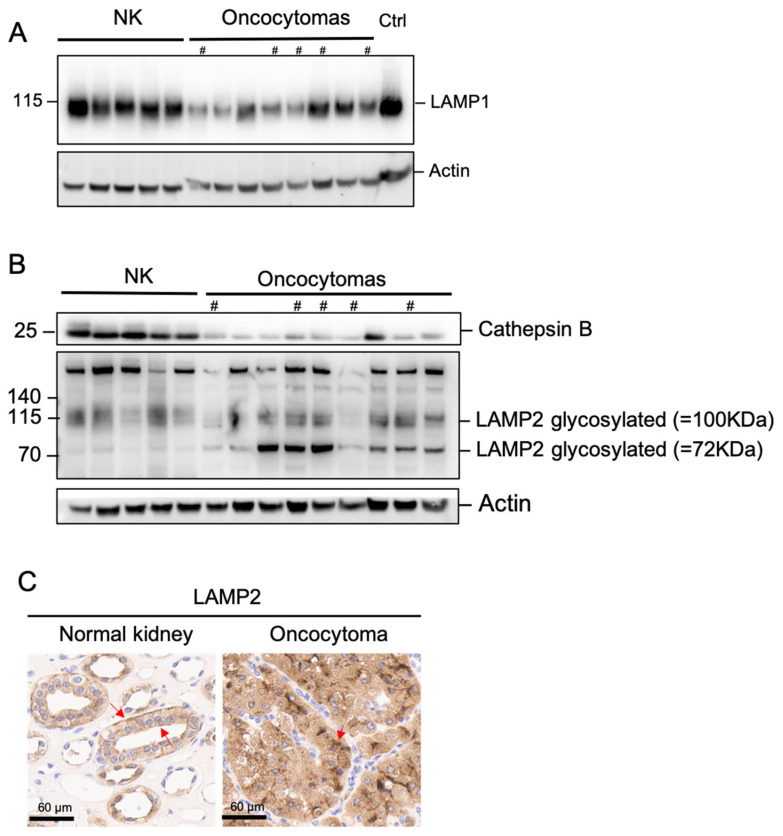
Decreased late-stage lysosome content in oncocytoma. (**A**) Western blot of LAMP1 in frozen oncocytoma and normal kidney tissues. (**B**) Western blot of cathepsin B and LAMP2 and its glycosylated forms in frozen oncocytoma and normal kidney tissues. (**C**) LAMP2 differential distribution in normal kidney and oncocytoma using IHC. Arrows indicate the subcellular locations of LAMP2 staining. # Samples with a high frequency of LoF Complex I mutations. The same loading control, actin, was used for Figure 3C,E and Figure 5A because the same membrane was used to immunoblot these proteins.

**Figure 6 ijms-26-07654-f006:**
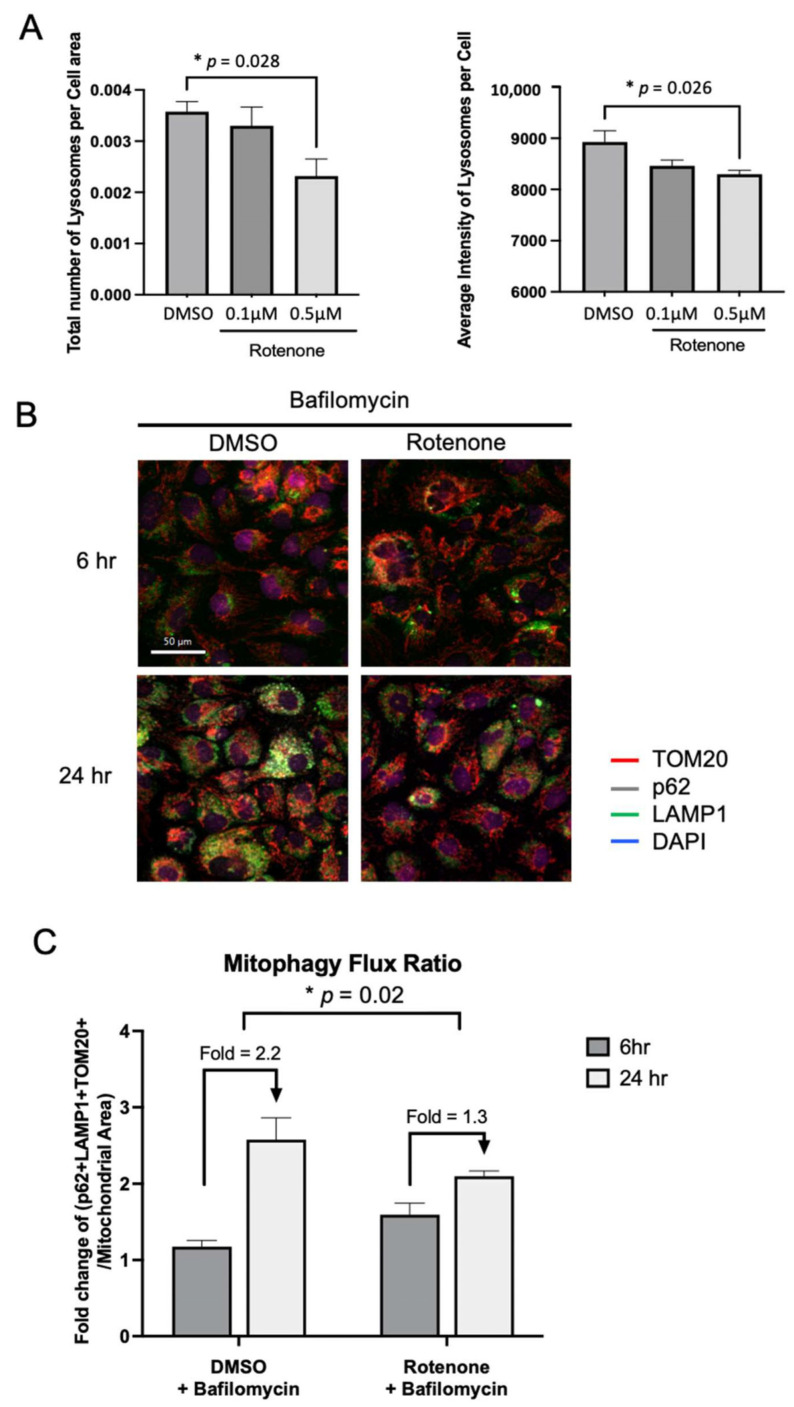
Inhibition of Complex I using rotenone decreases lysosome content and impairs mitophagy flux in YUNK1. (**A**) Quantification of lysosome using lysotracker in YUNK1 cell after treatment with rotenone 0.1 μM or 0.5 μM, or DMSO for 24 h. (**B**) Immunofluorescence of mitolysosome using co-staining of TOM20 (red), p62 (grey), LAMP1 (green), and DAPI (blue). (**C**) Mitophagy flux of YUNK1 was calculated as the fold change from treatment with rotenone or DMSO in the presence of bafilomycin compared to in the absence of bafilomycin, at 6 and 24 h time points. *P*-value was generated by comparing fold changes between 6 h and 24 h of treatment. * *p* < 0.05.

**Figure 7 ijms-26-07654-f007:**
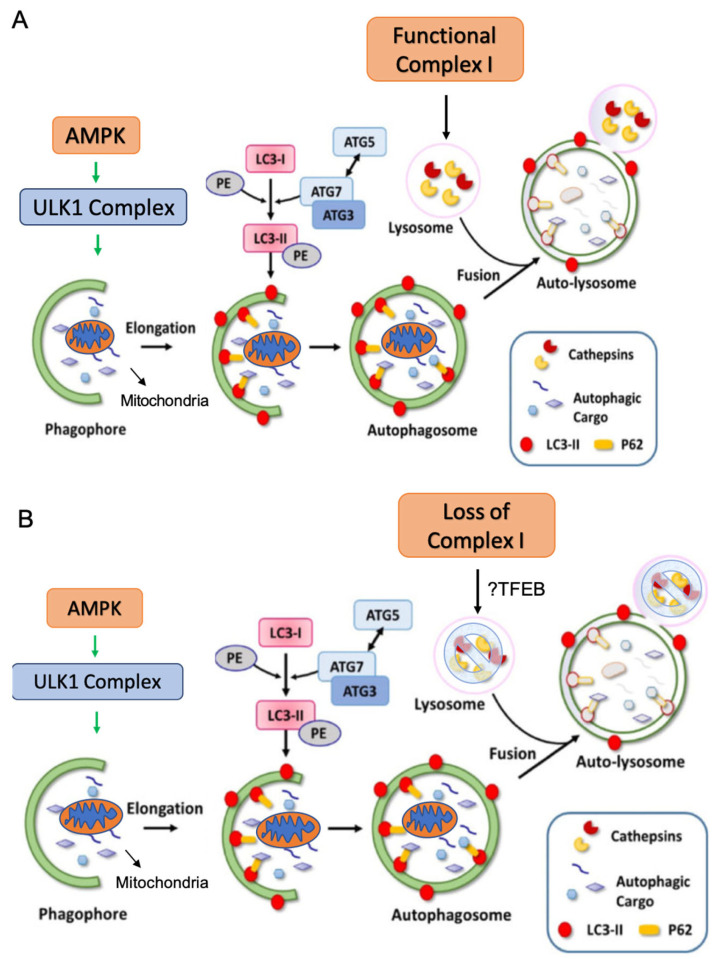
Illustration of mechanism of Complex I loss in oncocytoma, resulting in defective mitophagy via lysosomal dysfunction. (**A**) Competent Complex I under normal condition. (**B**) Complex I loss in renal oncocytoma. Adapted from [28].

## Data Availability

The data are contained within the article or Appendix A.

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
