# Peer review of "The Loss of Complex I in Renal Oncocytoma Is Associated with Defective Mitophagy Due to Lysosomal Dysfunction"

_ijms, 2025, doi:10.3390/ijms26157654_

Round 1

Reviewer 1 Report

Comments and Suggestions for Authors

The manuscript submitted to the IJMS journal by Lin et al. suggests a possible relationship between CI respiration deficiency (nonsense mutation in the mitochondrial DNA-encoded CI subunits) and defective autophagy (mitophagy) in benign renal oncocytomas. The renal oncocystomas do have downregulated expression of CI complex subunits and increased expression of CIV complex subunits and attenuated CI-linked respiration. These changes/modulations in mitochondrial respiration result in increased accumulation of mitochondria (likely to cope with suppressed mitochondrial respiration) and increased activation of autophagy/mitophagy in these cells. Interestingly, behind the accumulation of apparently less respiring mitochondria in these cells is a lower level/activity of the lysosomal proteins. Attenuated levels of the lysosomes/main lysosomal proteins, such as cathepsin B can result from suppressed activity of the transcription factor TFEB (possibly via increased AMPK activity), which positively regulates the expression of many lysosomal genes. The authors also documented that forced inhibition of CI-linked respiration by rotenone could lead to decreased lysosomal content and thus likely attenuated mitophagy. However, as the authors also admit, there is no solid proof that decreased lysosomal content is a direct outcome of attenuated CI-linked respiration.

Questions and comments:

  1. The authors claim that they high resolution respirometry via Seahorse for the determination of CI- and CIV-linked respiration of cell lysates, and in the methodical part refer to 2 publications – cit. 27 and 28. First of all, there is no description of the used protocol for the CI and CIV-linked respiration and those two references do not contain relevant methodical data (in contrast to e.g. Osto et al.  https://doi.org/10.1002/cpcb.116 ). Thus, the relevant protocol and also the primary data (not just their graphical interpretation) must be presented.
  2. How do the authors interpret the increased levels (and respiration) of the CIV and likely CIII complex? Increase mitochondrial content… ? Unlikely as the CII (SDHB) is almost unchanged (or is it downregulated as well?).
  3. Autophagy regulators p62 and ULK1 are upregulated only in about half of RO – any explanation? What is the control in Fig. 3E? Are the lysosomal genes as cathepsin B or Lamp1 also downregulated at the transcription level - please analyze.
  4. What is the origin of YUNK1 cells, and is the 4.2. section in M&M related to them?
  5. In addition to rotenone, it would be instrumental to downregulate the expression of a CI nuDNA-encoded gene (e.g. Ndufa9 or other – Ndufs1, …) using shRNA in YUNK1 or eventually, even HEK293 cells and determine the impact of this downregulation on mitochondrial respiration and the lysosomal content.

Author Response

Summary

The manuscript submitted to the IJMS journal by Lin et al. suggests a possible relationship between CI respiration deficiency (nonsense mutation in the mitochondrial DNA-encoded CI subunits) and defective autophagy (mitophagy) in benign renal oncocytomas. The renal oncocystomas do have downregulated expression of CI complex subunits and increased expression of CIV complex subunits and attenuated CI-linked respiration. These changes/modulations in mitochondrial respiration result in increased accumulation of mitochondria (likely to cope with suppressed mitochondrial respiration) and increased activation of autophagy/mitophagy in these cells. Interestingly, behind the accumulation of apparently less respiring mitochondria in these cells is a lower level/activity of the lysosomal proteins. Attenuated levels of the lysosomes/main lysosomal proteins, such as cathepsin B can result from suppressed activity of the transcription factor TFEB (possibly via increased AMPK activity), which positively regulates the expression of many lysosomal genes. The authors also documented that forced inhibition of CI-linked respiration by rotenone could lead to decreased lysosomal content and thus likely attenuated mitophagy. However, as the authors also admit, there is no solid proof that decreased lysosomal content is a direct outcome of attenuated CI-linked respiration.

Response: Thank you very much for taking the time to review this manuscript. Please find the detailed responses below and the corresponding revisions/corrections highlighted/in track changes in the re-submitted files.

Comments 1: The authors claim that they high resolution respirometry via Seahorse for the determination of CI- and CIV-linked respiration of cell lysates, and in the methodical part refer to 2 publications – cit. 27 and 28. First of all, there is no description of the used protocol for the CI and CIV-linked respiration and those two references do not contain relevant methodical data (in contrast to e.g. Osto et al. https://doi.org/10.1002/cpcb.116 ). Thus, the relevant protocol and also the primary data (not just their graphical interpretation) must be presented.]

Response 1: Thank you for pointing this important oversight on our end.   The citations were off, we meant to cite the Osto et al to better describe the methodology. We have now fixed the references and cited the appropriate work including the particular changes in line 424.

Comments 2: How do the authors interpret the increased levels (and respiration) of the CIV and likely CIII complex? Increase mitochondrial content… ? Unlikely as the CII (SDHB) is almost unchanged (or is it downregulated as well?).

Response 2: We agree with the assessment by the reviewer that Complex III and IV are upregulated in oncocytomas probably due to the compensatory increase of mitochondrial content in oncocytomas. Unlike Complex III and IV which are both encoded by mitochondrial genes, Complex II is entirely encoded by nuclear genes which makes it less directly affected by the increase of mitochondrial content in oncocytomas. 1

Comments 3: Autophagy regulators p62 and ULK1 are upregulated only in about half of RO – any explanation? What is the control in Fig. 3E? Are the lysosomal genes as cathepsin B or Lamp1 also downregulated at the transcription level - please analyze.

Response 3: Thank you for the valuable comments. We are not entirely sure either but can speculate. P62 upregulation seems to be more prominent in oncocytomas with high variant allele frequency and higher mitochondrial content compared to the rest (Figure 3D and E). It could be that lysosome functions are more of a bottle neck in high mitochondrial specimens that p62 accumulates more. While p-ULK1 is higher in specimens with higher p-AMPK (Figure 4A), which is relatively about half the specimens, yet we don’t know why those specific specimens have higher AMPK activation.   Control in 3E control is mouse heart homogenates as a positive control.  We looked at the transcription of cathepsin A, cathepsin D and cathepsin L in normal kidneys vs oncocytomas using nCounter Metabolic panel, there are no significant differences in these gene transcripts. We did not specifically look at the transcription of cathepsin B or LAMP1, but it is something that can be done in the future.

Comments 4:[What is the origin of YUNK1 cells, and is the 4.2. section in M&M related to them?.

Response 4: Thank you for pointing out the lack of clarity here. YUNK1 is a SV40-immortalized Yale University normal kidney cell line, additional characterization of this cell line was described previously and was derived from normal renal cortical tissue.2 We also edited the main text, see line 197-198.

Comments 5: In addition to rotenone, it would be instrumental to downregulate the expression of a CI nuDNA-encoded gene (e.g. Ndufa9 or other – Ndufs1, …) using shRNA in YUNK1 or eventually, even HEK293 cells and determine the impact of this downregulation on mitochondrial respiration and the lysosomal content.

Response 5: This is a very insightful question. We have had the ultimate goal of knocking down a subunit of Complex I to look at consequent mitochondrial respiration and if it recapitulates oncocytoma biology for many years. Using shRNA to knockdown Ndufa9 is a very reasonable approach to answer this question, however there are some limitations with this technique including the lack of complete knockdown and potential off-target effects. Stroud et al, tested the feasibility of Transcription activator-like effector nucleases (TALEN)-mediated gene disruption of Ndufa9 to study complex I biogenesis in HEK294 cells. They found homozygous disruption of the Ndufa9 gene resulted in loss of complex I activity and found that NDUFA9 is important in stabilizing the junction between matrix and membrane arms of complex I.3 Instead of targeting a complex I nuclear subunit, we have created the knockout of mitochondrial-encoded subunit nd4 using the novel mitoTALEN.4 We have successfully obtained heteroplasmy of nd4 in YUNK1 cell line, however these cells are very slow growing but with rapid loss of nd4 heteroplasmy over time/passages. We are still in the process of optimizing the mitoTALEN protocols hoping to achieve a sufficiently stable cell line with high mutant allele fraction to investigate downstream effects of complex I loss in the future.   We also created a patient-derived cell line which had a somatic MT-ND5 mutation (with near homoplasmy (>80% mutation frequency)), and demonstrated a loss of ND5 expression, loss of complex I activity, a high content of mitochondrial DNA, and a near loss of oxidative phosphorylation and reliance on aerobic glycolysis. However, this cell line was very slow growing and we lost the cell line due to contamination. We hope in a future analysis we can show how genetic modification can lead to oncocytogenesis to prove that complex I dysfunction leads to the resultant biology we see in this manuscript.

References:

  1. Kremer LS, Rehling P. Coordinating mitochondrial translation with assembly of the OXPHOS complexes. Hum Mol Genet. 2024;33(1 R):R47-R52. doi:10.1093/hmg/ddae025
  2. Sulkowski PL, Sundaram RK, Oeck S, et al. Krebs-cycle-deficient hereditary cancer syndromes are defined by defects in homologous-recombination DNA repair. Nat Genet. 2018;50(8):1086-1092. doi:10.1038/s41588-018-0170-4
  3. Stroud DA, Formosa LE, Wijeyeratne XW, Nguyen TN, Ryan MT. Gene knockout using transcription activator-like effector nucleases (TALENs) reveals that human ndufa9 protein is essential for stabilizing the junction between membrane and matrix arms of complex i. Journal of Biological Chemistry. 2013;288(3):1685-1690. doi:10.1074/jbc.C112.436766
  4. Bacman SR, Kauppila JHK, Pereira C V., et al. MitoTALEN reduces mutant mtDNA load and restores tRNAAla levels in a mouse model of heteroplasmic mtDNA mutation. Nat Med. 2018;24(11):1696-1700. doi:10.1038/s41591-018-0166-8

Reviewer 2 Report

Comments and Suggestions for Authors

The author claimed that renal oncocytoma (RO) exhibited reduced Complex I expression and activity, and concluded that Complex I loss in RO results in defective mitophagy due to lysosomal loss and dysfunction. I have many concern in the data quality of this manuscript as shown below.

  1. The sentence in the title is not smooth,
  2. The Actin in Figure 5A is similar to the actin in Figure 3E and 3C. The last band (right side) in Figure 3E is Ctrl, but it doesn’t in Figure 5A. In addition, why the molecular weight of actin in oncocytomas is higher than that in NK? Again, the Actin in Figure 3B is similar to 4C
  1. There is a lack of summary graph to better improve out understanding of this manuscript.
  2. Figure 3D, IHC of LC3 in oncocytoma and its adjacent normal kidney is needed.
  3. Why the loading control in Figure 4A is GAPDH, rather than Actin.
  4. The staining of TOM20 (Red), p62 (Gray), LAMP1 (Green), and DAPI (Blue) should be labeled in figure.

Author Response

Summary

The author claimed that renal oncocytoma (RO) exhibited reduced Complex I expression and activity, and concluded that Complex I loss in RO results in defective mitophagy due to lysosomal loss and dysfunction. I have many concern in the data quality of this manuscript as shown below.

Response: Thank you very much for taking the time to review this manuscript. Please find the detailed responses below and the corresponding revisions/corrections highlighted/in track changes in the re-submitted files.

Comments 1: The sentence in the title is not smooth

Response 1: Thank you for pointing this out. We will change the title to flow better to:  “The Loss of Complex I in Renal Oncocytoma Is Associated with Defective Mitophagy due to Lysosomal Dysfunction.” See updated title.

Comments 2: The Actin in Figure 5A is similar to the actin in Figure 3E and 3C. The last band (right side) in Figure 3E is Ctrl, but it doesn’t in Figure 5A. In addition, why the molecular weight of actin in oncocytomas is higher than that in NK? Again, the Actin in Figure 3B is similar to 4C

Response 2: Figure 3E and 3C were run in the same experiment, Figure 5C was run as a separate experiment. 3C has a loading control as well but was cropped out. We have provided the entire uncropped WB for review. Actin as loading control was pretty consistent throughout, so they looked similar between the different panels. The molecular weight of the actin in oncocytomas vs normal kidney (NK) is probably due to an artifact which are quite common. 

Comments 3: There is a lack of summary graph to better improve our understanding of this manuscript.

Response 3: Thank you for pointing this out, we have provided a summary cartoon to help outline the key findings in the manuscript so it’s easier to follow. See Figure 7, and cited in discussion line 335.

Comments 4: Figure 3D, IHC of LC3 in oncocytoma and its adjacent normal kidney is needed.

Response 4: Thank you for pointing this out. We have tried to stain LC3 using Rabbit mAb (cell signaling #12741) for IHC, however, we had no success of detecting LC3 in our control specimens including normal kidneys and lung cancer specimens. We also tried to purchase mAB anti-LC3B from nanoTools (0231-100/LC3-5F10) per Joshi et al, however, we couldn’t buy that particular clone at this time due to current financial restrictions on essential purchases at our institution.  We would love to try demonstrate this via IHC and gave it our best effort with the available tools.

Comments 5: Why the loading control in Figure 4A is GAPDH, rather than Actin.

Response 5: GAPDH is a commonly used loading control that we are familiar with.  The Actin band was just at the cutting line when we cut the membrane bands to do stain for signaling proteins. GAPDH is a little bit lower so it was more feasible to use this instead.

Comments 6: The staining of TOM20 (Red), p62 (Gray), LAMP1 (Green), and DAPI (Blue) should be labeled in figure.

Response 6: Thank you for pointing this out. We can label that in the figure. See updated figure 6B.

Round 2

Reviewer 1 Report

Comments and Suggestions for Authors

In the authors response to to my 1st comment I still miss the primary data (OCR curves) and the exact methodology, not just corrected reference - both can be presented in the supplementary data. Similarly in the response to the 3rd comment I specifically asked for RT-qPCR analysis of cathepsin B or LAMP1 transcripts, which also was not by the authors provided...

Author Response

Comment: In the authors response to to my 1st comment I still miss the primary data (OCR curves) and the exact methodology, not just corrected reference - both can be presented in the supplementary data. Similarly in the response to the 3rd comment I specifically asked for RT-qPCR analysis of cathepsin B or LAMP1 transcripts, which also was not by the authors provided...

Response: Thank you for pointing this out- we provided clarification to address this concern. We have incorporated the primary OCR curves into our manuscript, see Supplemental Figure S1A. We picked 3 representative tracings from each group to show here, as it was not really feasible to visualize all the samples on the same graph. We also provided the exact methodology for respirometry in frozen samples, see supplemental methodology. Corresponding updates were made in main text, see line 107- 109, 435. See the attachment of the updated supplemental data and methods.

We would be happy to perform RT-qPCR for cathepsin B and LAMP1 transcripts in our normal kidney and oncocytoma cohorts. There, however, is a freeze on spending right now at UCLA due to massive budget cuts due to grant cancellation and federal indirect cost policies. We can order things and get approved as this would be considered essential to the research, but it would take up to 30 days for a review. While we do not think the data will change our conclusions, if needed, we just need up to 90 days for an extension.

Reviewer 2 Report

Comments and Suggestions for Authors

The data quality in this work is low, in particularly the western blot were similar in some figures. I thus suggest to reject this manuscript. 

Author Response

Comment: The data quality in this work is low, in particularly the western blot were similar in some figures. I thus suggest to reject this manuscript.

Response: We would like to clarify our experimental workflow to address the reviewer's concerns. Our approach is common when tumor samples are limited with a workflow that maximizes the data obtained from every valuable sample’s immunoblot. Normal kidneys and oncocytomas are always together in the same PVDF membrane to perform the immunoblot of a target protein. However, a single membrane is used to detect multiple proteins by either: (1) taking advantage of the different sizes of multiple targets and cutting the membrane horizontally; or (2) by performing stripping and reblotting. In each membrane we ran, we also immunoblotted for a housekeeping protein as a loading control. ACTIN, GAPDH or VINCULIN have been used for this purpose in the manuscript as reliable loading control proteins. Because several target proteins have been detected in the same membrane, their corresponding ACTIN is necessarily identical. After cropping and re-assembling panels to highlight different proteins during the manuscript, we needed to sometimes show the same ACTIN image. Our use of a single loading control per membrane is a widely accepted practice to ensure quantitative consistency when probing multiple targets sequentially. For transparency, we indicated similar images in the figure legends in the revised text, see figure legend 1, 3, 4 and 5.

  1. Figure 3C, Figure 3E and Figure 5A share the same loading control

  2. Figure 1E, Figure 3B, and Figure 4C share the same loading control

We noticed a mislabeling in Figure 5A in the previous version, so we corrected it. We also merged the lower panel of the previous figure 5A with 5B since they were from the same membrane and shared the same loading control.

Additionally, we have provided (since the initial submission) the uncropped images of all the blots to the editors, but we're afraid the reviewer may not have received that file and insufficient information was interpreted as evidence of poor data quality. The uncropped and annotated WB images are included here as an attachment, and we highlighted the figures sharing the same loading controls, as well as indicated the exact membrane that was used for each Western Blot. 

Round 3

Reviewer 1 Report

Comments and Suggestions for Authors

The authors at least fulfilled one of my main comments/objections on the data presentation and methodics of Seahorse-assisted respirometry, fulfilling the other one (mRNA data) is not as essential.